# Switching of Receptor Binding Poses between Closely Related Enteroviruses

**DOI:** 10.3390/v14122625

**Published:** 2022-11-24

**Authors:** Daming Zhou, Ling Qin, Helen M. E. Duyvesteyn, Yuguang Zhao, Tzou-Yien Lin, Elizabeth E. Fry, Jingshan Ren, Kuan-Ying A. Huang, David I. Stuart

**Affiliations:** 1Division of Structural Biology, Nuffield Department of Medicine, University of Oxford, The Wellcome Centre for Human Genetics, Headington, Oxford OX3 7BN, UK; 2Chinese Academy of Medical Sciences Oxford Institute, University of Oxford, Oxford OX3 7FZ, UK; 3Division of Pediatric Infectious Diseases, Department of Pediatrics, Chang Gung Memorial Hospital, Taoyuan 333, Taiwan; 4Graduate Institute of Immunology and Department of Pediatrics, National Taiwan University Hospital, College of Medicine, National Taiwan University, Taipei 100, Taiwan; 5Diamond Light Source Ltd., Harwell Science & Innovation Campus, Didcot OX11 0DE, UK

**Keywords:** echovirus E11, virus receptor, DAF, complex, binding pose, glycan, evolution, enterovirus structure

## Abstract

Echoviruses, for which there are currently no approved vaccines or drugs, are responsible for a range of human diseases, for example echovirus 11 (E11) is a major cause of serious neonatal morbidity and mortality. Decay-accelerating factor (DAF, also known as CD55) is an attachment receptor for E11. Here, we report the structure of the complex of E11 and the full-length ectodomain of DAF (short consensus repeats, SCRs, 1–4) at 3.1 Å determined by cryo-electron microscopy (cryo-EM). SCRs 3 and 4 of DAF interact with E11 at the southern rim of the canyon via the VP2 EF and VP3 BC loops. We also observe an unexpected interaction between the N-linked glycan (residue 95 of DAF) and the VP2 BC loop of E11. DAF is a receptor for at least 20 enteroviruses and we classify its binding patterns from reported DAF/virus complexes into two distinct positions and orientations, named as E6 and E11 poses. Whilst 60 DAF molecules can attach to the virion in the E6 pose, no more than 30 can attach to E11 due to steric restrictions. Analysis of the distinct modes of interaction and structure and sequence-based phylogenies suggests that the two modes evolved independently, with the E6 mode likely found earlier.

## 1. Introduction

Echoviruses are responsible for a range of human diseases, usually mild, like hand, foot and mouth disease, but sometimes severe including aseptic meningitis, myocarditis and encephalitis [1,2,3]. Of all types of echoviruses, E11 is the most frequent cause of serious neonatal disease and death [4]. Currently, there are no licensed drugs for echovirus infection and no available vaccines.

Echoviruses, non-enveloped viruses with a positive single-stranded RNA genome of ~7500 bp, belong to the species *enterovirus B* of the genus *Enterovirus*, the most populous genus of the family *Picornaviridae*. The capsid consists of 60 identical protomers arranged with icosahedral symmetry and each protomer is composed of four capsid proteins, VP1–4, derived from the P1 polyprotein. VP1–3 all have a core structure called a ‘jelly roll’ and form the outer surface of the capsid, whilst VP4 is located on the inner surface of the capsid. Enteroviruses are characterized by a depression called the “canyon”, encircling the 5-fold axes of symmetry of the capsid, that was initially proposed to be the binding site of IgG-like picornavirus receptors [5]. A number of uncoating receptors were subsequently confirmed to bind the canyon [6,7] although this is not universal [8]. In echoviruses the canyon is not continuous (as is the case with species A and C enteroviruses, which include polioviruses, a range of Coxsackie viruses and Enterovirus A71 (EV71)) but is split into five pits where the uncoating receptor (neonatal Fc receptor or FcRn) binds [9]. Below the floor of the enterovirus canyon a lipid molecule, termed pocket factor, binds VP1 inside a hydrophobic pocket and is generally expelled as a prelude to particle expansion and viral uncoating early in infection.

DAF is one of the regulators of the complement system, inhibiting formation of the complement convertases in both the classical and alternative pathways [6,7,10,11,12]. DAF also functions in signal transduction in T cells [13]. It is a membrane glycoprotein, widely distributed on the surface of all types of cells that are in contact with plasma complement proteins and attached to the lipid bilayer by a C-terminal glycosylphosphatidylinositol (GPI) anchor. The extracellular region of DAF is composed of four short consensus repeats (SCR1-4) [14,15]. Many echoviruses have been shown to use DAF as an attachment receptor [16,17] and several structures of enteroviruses in complex with DAF have been published [9,18,19,20,21].

Since DAF has been seen to bind in different ways to different enteroviruses and there is substantial divergence in the sequence of P1 between E11 and other DAF-binding enteroviruses, we determined the structure of E11 in complex with DAF at a resolution of 3.1 Å to establish the precise mode of interaction. This structure reveals that SCR 3 and 4 of DAF and the VP2 EF loop and VP3 BC loop of E11 are involved in the interaction and an N-linked glycan on DAF also interacts with the E11 capsid. We were able to classify the modes of DAF binding into two distinct poses which we propose evolved independently.

## 2. Methods

### 2.1. Expression and Purification of DAF

The cDNA of human DAF (UniProtKB P08174) was ordered from Horizon Discovery Ltd, Cambridge, UK (cDNA clone ID: 3460621). The DNA encoding SCR 1-4 of DAF (residues 35-285 aa) was amplified by PCR with primers 5′-TGTTACCGGTGACTGTGGCCTTCCCCCAGAT -3′ (forward) and 5′-ACTTGGTACCTCCTCTGCATTCAGGTGGTGGGC (reverse). Amplified cDNA was cloned into the pHL-Sec vector, with sequences encoding a C-terminal 6*His tag added. HEK293T cells (ATCC, Manassas, VA, USA CRL-11268) were transfected with the recombinant plasmid pHC-DAF. The conditioned medium was harvested and dialysed for 2 days in 23.2 mM Na_2_HPO_4_, 1.7 mM NaH_2_PO_4_, pH 8.0, 250 mM NaCl at 4 °C. DAF extracellular domain was purified with a 5 mL nickel column (GE, Boston, MA, USA) and polished using a Superdex 75 Hiload 16/60 gel filtration column (GE), eluted in 20 mM Tris, 200 mM NaCl, pH 7.4. Purified protein was concentrated using a 10 KD centrifugal filter (Sigma-Aldrich, Burlington, MA, USA).

### 2.2. Virus Production and Purification

E11 (strain 18-2135) was used to infect human Rhabdomyosarcoma (RD) cells. 4 days after infection, virus was harvested and precipitated by adding 8% (*w*/*v*) PEG 6000 and 0.5% (*v*/*v*) NP40. The sample was centrifuged at 3500× *g* for 1 h at 4 °C and the pellet stored at −80 °C until required. The thawed pellet was suspended in buffer (50 mM HEPES, pH 7.4, 200 mM NaCl, 0.5% (*v*/*v*) NP40) and centrifuged at 3500× *g* for 30 min at 4 °C to remove cell debris. Virus particles in the supernatant were pelleted through a 2 mL 30% (*w*/*v*) sucrose cushion (in 50 mM HEPES, pH 7.4, 200 mM NaCl) at 154,000× *g* for 3 h at 4 °C using a SW32 rotor (Beckman). The pellet was suspended in buffer (50 mM HEPES, pH 7.4, 200 mM NaCl, 0.5% (*v*/*v*) NP40) and centrifuged at 12,000× *g* for 30 min at 4 °C twice to remove insoluble material. The supernatant was then layered on the top of a 15–45% sucrose gradient (in 50 mM HEPES, pH 7.4, 200 mM NaCl) and centrifuged at 154,000× *g* for 3 h at 4 °C using a SW32 rotor (Beckman, Brea, CA, USA). ~10 mL of fractions containing E11 full particles were harvested and diluted in buffer (50 mM HEPES, pH 7.4, 200 mM NaCl). The sample was centrifuged at 154,000× *g* for 3 h at 4 °C again to pellet virus particles using a SW32 rotor (Beckman). Then, the supernatant was discarded, and the pellet dissolved in buffer (50 mM HEPES, pH 7.4, 200 mM NaCl). The concentration of virus particles was measured using a NanoDrop™ spectrophotometer. 

### 2.3. Cryo-EM Grid Preparation

E11 particles (~2 mg/mL) were incubated with DAF protein (4.3 mg/mL) at room temperature for 1.5 h, with a molar ratio of DAF:E11 protomer of 3:1 (i.e., 180 DAF molecules per virus particle). 4 μL of the sample was applied to a glow-discharged ultra-thin carbon Lacey grid (Agar Scientific,, London, UK S187-4), blotted by filter paper from the back of the grid (blotting time ~3 s) and vitrified by plunging into liquid ethane/propane mix (50%:50%) using a manual plunger (Max Planck Institute of Biochemistry, Munich, Germany).

### 2.4. Cryo-EM Data Collection

Cryo-EM data were collected using a 300-kV Titan Krios microscope (Thermo Fisher Scientific, Waltham, MA, USA), equipped with a Falcon III detector (FEI) and modified to operate with infectious particles (Table 1). Data were recorded as movies (24 frames, each 0.25 s) in linear mode using the EPU software (FEI) with a defocus range of −1 to −3 μm. The magnification was nominally 130,000×, corresponding to a calibrated pixel size of 1.05 Å. The dose rate was ~7 e^-^/Å^2^/s, resulting in a total electron dose of 41 e^-^/Å^2^.

### 2.5. Cryo-EM Data Processing

Frames of each movie were aligned and averaged using MotionCor2 (version 1.3.0, Shawn Zheng, University of California, San Francisco, CA, USA) [22] and contrast transfer function (CTF) parameters were determined with CTFFIND 4 (version 4.0.7, The Grigorieff lab, University of Massachusetts Chan Medical School, Worcester, MA, USA) [23]. Micrographs with significant drift or astigmatism were discarded. Particles were automatically picked using Warp (version 1.0.6, Dimitry Tegunov, Max Planck Institute for Biophysical Chemistry, Göttingen, Germany) [24]. The structure was calculated with Relion 3.1.4 (Sjors Scheres, MRC Laboratory of Molecular Biology, Cambridge, UK) following the gold-standard refinement procedure [25]. Reference-free 2D-class averaging was performed. Model-based 3D classification and refinement was run using initial models generated by filtering the crystal structure of echovirus 11 (PDB: 1H8T) [26] to 50 Å resolution. Post-processing and CTF refinement were run in Relion to further improve the resolution of the density map. The final density map was calculated using 6394 particles from 6469 micrographs, with an overall resolution of 3.1 Å, estimated by Fourier shell correlation at 0.143 [27]. 

To do symmetry expansion, cryo-EM data were re-processed using Cryosparc (version 3.3.2, Structura Biotechnology Inc, Toronto, Canada). Patch motion correction (multi) and CTF estimation using Gctf BETA (version 1.06, Kai Zhang, Yale University, New Haven, CT, USA) were applied to 7088 micrographs. Particles were picked by blob picker, extracted with 42 nm boxes and selected by 2D classification. 10,479 particles were subjected to Non-uniform Refinement (I1 symmetry), and the final result from Relion was used as an input volume. The particles were then expanded 60-fold by I1 symmetry expansion. A mask focusing on symmetrical density from two DAF molecules at an icosahedral 2-fold area was created from a fitted DAF model with 10 Å dilation radius and 3 Å soft padding width. An inverse mask was created for particle subtraction of the symmetry expanded particles. 3D variability analysis was applied, and 3 clusters were obtained, two of which showed one DAF and the other no DAF (corresponding to 31.5% DAF occupancy). Local refinement was performed on one of the clusters containing DAF (the other simply being symmetry related to this). After global and local CTF refinement, another round of local refinement was applied, and the final map was sharpened with B factor −181.5. See Table 1. The cryo-EM workflow is shown in Appendix A.

### 2.6. Model Building

The crystal structure of E11 strain EV11-207 (PDB: 1H8T) [26] and the structure of DAF (PDB: 1OK3) [15] were fitted into the electron potential map in COOT 0.9.8.1 (Paul Emsley, MRC Laboratory of Molecular Biology, Cambridge, UK) [28]. The amino acid sequence was corrected to E11 strain 18-2135. The pixel size of the map was calibrated to 1.05 Å by assuming the particle size of the crystal structure was exactly right. The model was further improved using Phenix Real-space-refine (version 1.19.2-4158, The Phenix development team, multi-country) [29]. Refinement statistics are given in Table 1. The residues forming the E11-DAF interface were identified with PISA 1.48 (European Bioinformatics Institute, Hinxton, UK) [30]. Roadmaps were calculated using Rivem 5.1 (Chuan Xiao, University of Texas, El Paso, TX, USA) [31]. All figures were prepared with PYMOL2.5.2 (Schrodinger, Inc., New York, NY, USA) [32], Chimera 1.15 (University of California, San Francisco, CA, USA) [33] and ChimeraX 1.5 (University of California, San Francisco, CA, USA) [34].

### 2.7. Phylogenetic Analysis

A structure-based phylogenetic tree was drawn using SHP (David Stuart, University of Oxford, Oxford, UK) [35] and a sequence-based tree using ETE3 3.1.2 (Jaime Huerta-Cepas, EMBL, Heidelberg, Germany) [36].

## 3. Results

### 3.1. E11-DAF Interactions

The cryo-EM structure of the E11-DAF complex at neutral pH (pH 7.4) was determined at a nominal gold-standard resolution of 3.1 Å (see Methods and Table 1). The density for the virus is of high quality (Figure 1a–c) and the structure closely resembles the crystal structure of isolated mature E11 (PDB: 1H8T) [26], RMSD in Cαs 0.4 Å, although the identity of amino acids between these two strains is only 90.3%. In contrast the density for the DAF was weak. Whilst at low resolution the molecular envelope was clear (Figure 1d,e), at high resolution the density became fragmentary at a contour level where the capsid density was still continuous. It was clear that this was in substantial part because the DAF bound close to the icosahedral 2-fold axes, so that at most only 30 DAFs could be bound per particle, in particular, SCR domains 2 and 3 of the 2-fold related DAF molecules clash (Figure 1f). In order to improve the density and characterize the interaction, symmetry expansion and selection of DAF occupied sites by variability analysis was performed (see Methods). This analysis also indicated that the average number of DAF molecules bound per particle was ~20 (Methods) and improved the density allowing, where there were contacts with the virus, some larger side chain orientations to be determined. Thus the variability analysis selected the occupied sites, producing density of sufficient quality that the virus-receptor interactions described below are secure. We find that the internal structures of the four SCR domains of DAF are closely similar to those of the crystal structure (PDB: 1OK3) [15], although the relative position of SCR 1 is significantly different, which may be caused by crystal packing in the isolated molecule (Figure 1g). As expected for engagement with an attachment receptor with relatively weak binding (reported KD for E11 ∼3.0 μm [37]), which is not responsible for internalisation and genome release [9], the E11 structure is in its un-expanded state with the natural pocket factor (modelled as sphingosine) bound in the VP1 pocket (Figure 1h).

DAF binds E11 at the southern rim of the canyon, rather than into the canyon itself. This binding pattern is reminiscent of that of SCARB2 binding to EV71 [8], however, SCARB2 is an uncoating receptor whilst DAF is not. The footprint of the DAF receptor on the virus is ~900 Å^2^ (Figure 2a). The VP2 EF loop of E11 and SCR 3 of DAF dominate the interaction, although the VP3 BC loop of E11 and DAF SCR 4 are also involved (Figure 2b,c). Detailed interactions are listed in Table 2. Density for the glycan at residue 95 of DAF SCR1 can be clearly seen in the cryo-EM map and this glycan makes an interaction with the capsid of E11 (Figure 2d), which was not observed in a previously published E11-DAF structure (PDB: 6LA5), since that DAF construct only included SCRs 3 and 4. The density for the glycan is not well enough defined to describe the interaction in detail, however it is clear that the BC loop of VP2 of E11 is involved, centered on Lys73 (Figure 2d). Glycans are frequently used as attachment receptors for viruses, and we note that for foot-and-mouth disease virus a glycan on the integrin receptor was observed to make an interaction with the virus capsid at the point where heparan sulphate also bound [38]. It is therefore interesting that heparan sulphate has also been found to act as an attachment receptor in some echoviruses, including some that also bind DAF [39].

### 3.2. Different Binding Patterns of Enteroviruses with DAF

At least 20 enteroviruses (E3, E6, E6′, E7, E11-13, 19-21, E24-25, E29-30, E33, Enterovirus 70 (EV70), Coxsackievirus A21 (CVA21), CVB1, 3 and 5) use DAF as an attachment receptor. We find that the reported structures of DAF with 6 different enteroviruses cluster into two distinct patterns of DAF binding which we term the E6 pose and the E11 pose (Figure 3a). The orientation of DAF is markedly different between these poses. In the first pose (E6, E30 and CVB3) DAF binds with SCRs 3 and 4 sitting above and interacting with mainly VP2 as the DAF runs across the southern rim of the canyon at a slight angle with SCR2 interacting with the VP1 & VP3 C-termini from a neighbouring protomer, (Figure 3a). The second pose is exemplified by our structure of E11-DAF and is shared by E7 and E12. In this second pose DAF SCRs 3 and 4 bind at the southern rim of the canyon but are displaced clockwise around the rim by almost the length of an SCR and are pivoted clockwise by approx. 60° relative to the E6 pose so that SCRs 3 and 4 run along the canyon rim interacting with VP2 and VP3, whilst SCRs 1 and 2 project outwards from the canyon. (Figure 3a).

In the E6 pose, SCRs 2, 3 and 4 of one DAF molecule and two protomers of the capsid are involved in the interaction whilst in the E11 pose, only SCRs 3 and 4 and one protomer are directly involved, with the glycan of SCR1 interacting with a second protomeric unit. As a result of the differences a maximum of 30 DAF molecules can attach to E11, due to steric clashes near the 2-fold axis, whereas 60 copies of DAF can attach to the capsid in the E6 pose.

A structure based phylogenetic tree shows that the capsid structures cluster in-line with DAF binding with E6, E30 and CVB3 forming one branch and E7, E11 and E12 the second (Figure 3b). To investigate the virion structure/sequence correlates with DAF binding we performed a sequence based phylogenetic analysis of the 20 identified DAF binding enteroviruses (Figure 3c). Whilst E7, 11 and 12 partition to a single branch, E6, E30 and CVB3 are spread across the tree, suggesting that the viruses with each binding pose of DAF have coherent and distinct evolutionary histories. It seems that the E6 binding mode is more extensive and more ancient, with E11 pose viruses arising from a single, later, switch (Figure 3c).

Close examination of the capsid sequences of the viruses in the two distinct DAF-binding clades shows that E11-pose viruses (E7, 11 and 12) have a glycine at residue 162 of VP2 while E6-pose viruses mostly have larger asparagine or aspartic acid residues at this position: VP2 162N in E6 and VP2 162D in E30, although CVB3 has VP2 162S. In support of this, it has been reported that a single amino acid substitution G162E in VP2 caused the loss of E11 affinity to DAF [40]. Our structure confirms that a longer side chain at this position would clash with DAF in the E11 pose (Figure 3d), so VP2 162 is a key residue controlling DAF binding pose. In addition, we note that further residues involved in binding, VP1 269/270, VP2 165, VP3 76 of E11-pose viruses and E19 are conserved (Figure 3c), suggesting a role in controlling the DAF binding pattern, and corroborating the inference from the phylogenetic analysis that E19 belongs to the E11-pose group (Figure 3e).

## 4. Discussion

Structures of enterovirus-DAF complexes reveal two distinct DAF binding patterns which we refer to as the E6 and E11 poses, with the complex structures observed for six different enteroviruses partitioning equally between the two groups. The E11 pose we observe has been corroborated by mutational analysis [40]. Taking our structure of E11/full-length DAF together with previously reported results suggests that VP2 162 is important in determining the pose adopted by DAF. Although this diversity of poses for a common receptor on related viruses might appear surprising it is known that whilst the underlying mechanism of virion assembly and hence overall capsid structure is highly conserved, the surface of the virion, which modulates antigenicity and interactions with cell surface receptors, can evolve quickly [41] allowing viruses to modify tissue tropism. It is indeed likely that escape from antibody neutralization contributes to more rapid receptor diversification. In line with this, different enteroviruses use a wide variety of receptors. DAF-binding enteroviruses use DAF as an attachment receptor whilst other receptors, typically FcRn, trigger capsid expansion and uncoating. In line with this DAF binding is relatively low affinity for both poses, being reported as ~3 μM for E11 and ~0.4 μM for E6 [9,37]. The two DAF poses we describe are quite different, with no common interactions between the two. It might therefore be supposed that DAF binding evolved independently in each clade, presumably conferring increased infectivity and a selective advantage. Taking the structural information with the sequence-based phylogeny (Figure 3b,c) suggests indeed that the two clades are distinct, however it also suggests that the E6 pose was discovered first and is still the most common pose, whereas the E11 pose arose at some point through a switch in binding mode, since although there are no common interactions between the two poses, they cannot be occupied at the same time (Figure 3a). It is conceivable that the addition of the glycan interaction conferred a slight advantage, enough to fix the E11 binding pose in the population and create a new virus species, although as noted above the binding is reported to be relatively low affinity for both poses. In summary, it is clear that the functional role of cell attachment can be fulfilled by a variety of binding modes, and even specific protein–protein binding modes can evolve relatively easily, mirroring the facility with which many picornaviruses can adapt to tissue culture by acquiring the ability to bind cellular glycans, e.g., heparan sulphate [42,43]. Once acquired such binding will then act as a constraint (unless supplanted by a more effective interaction), leaving an evolutionary footprint on both the structure and sequence of the virus, which is reflected in the clean separation of the DAF poses into two clades on the basis of the virion structure (Figure 3b).

## Figures and Tables

**Figure 1 viruses-14-02625-f001:**
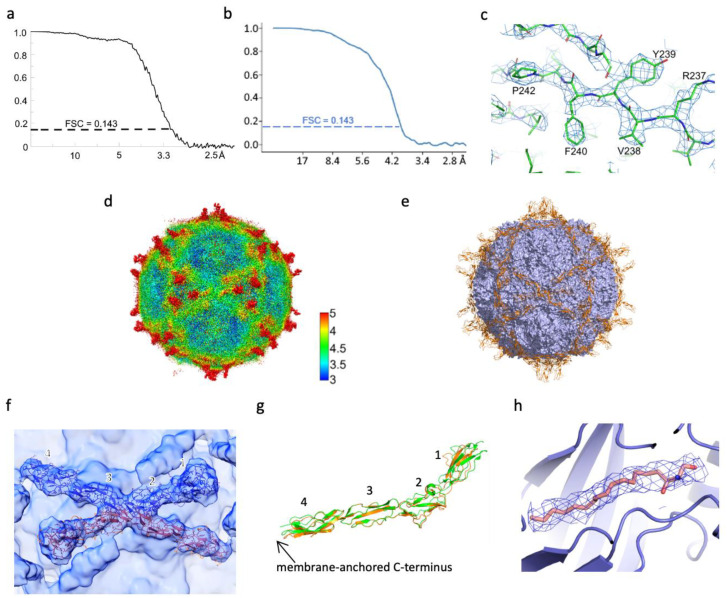
Overall structure of E11-DAF complex. (**a**) FSC curve of the E11-DAF complex with the threshold 0.143, the final resolution is 3.1 Å (calculated by Relion). (**b**) FSC curve of the local refined map centred at DAF with the threshold 0.143, the overall resolution is 3.9 Å (calculated by Cryosparc). (**c**) Representative density of E11 viral capsid. (**d**) Local resolution map of E11-DAF complex. (**e**) Three-dimensional reconstruction of E11-DAF complex with fitted DAF shown as cartoon. (**f**) Close-up of the Cryo-EM reconstruction of the E11-DAF complex, showing density for two DAF molecules (orange and red) related by icosahedral 2-fold. Note, domains SCR 2 and 3 of these two DAF molecules clash with each other. (**g**) Superimposition of the E11 bound DAF (orange) with that determined by crystallography (green, PDB: 1OK3). (**h**) Density of the pocket factor.

**Figure 2 viruses-14-02625-f002:**
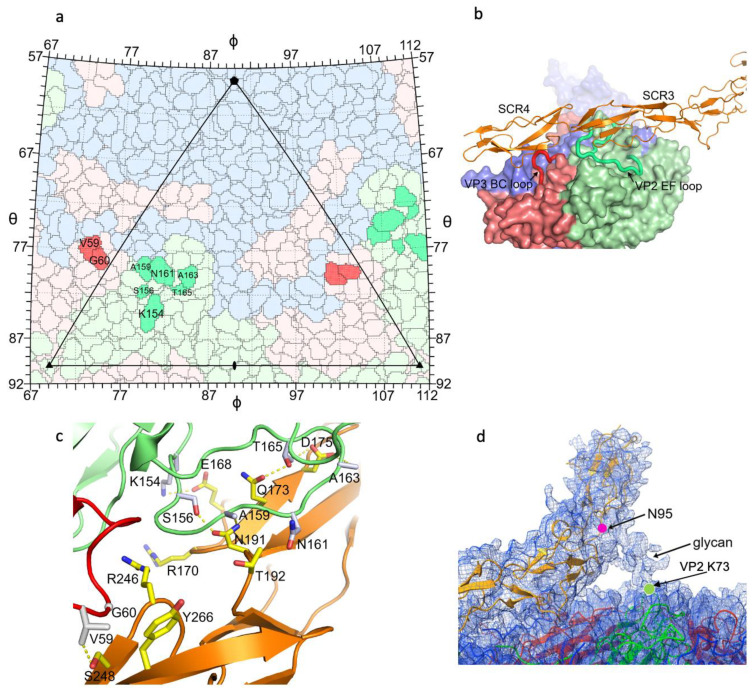
Interactions of E11 and DAF. (**a**) Roadmap [31] showing the residues of E11 involved in E11-DAF protein interactions. (**b**) SCRs 3 and 4 of DAF (orange), and the VP2 EF loop (green)and VP3 BC loop (red) of E11 are involved in E11-DAF interactions. The viral protomer is drawn as surface representation with the two loops involved in contacts with DAF shown as coils. VP1, 2 and 3 of E11 are colored in blue, green and red, respectively. (**c**) Binding residues at the E11-DAF interface (E11: VP2 green, VP3 red, DAF orange). Side chains of E11 are shown as grey sticks and those of DAF shown as yellow sticks; (**d**) Density showing contact between the Glycan attached to residue N95 of DAF (shown as a magenta circle) and residue K73 of VP2 (marked with a green circle).

**Figure 3 viruses-14-02625-f003:**
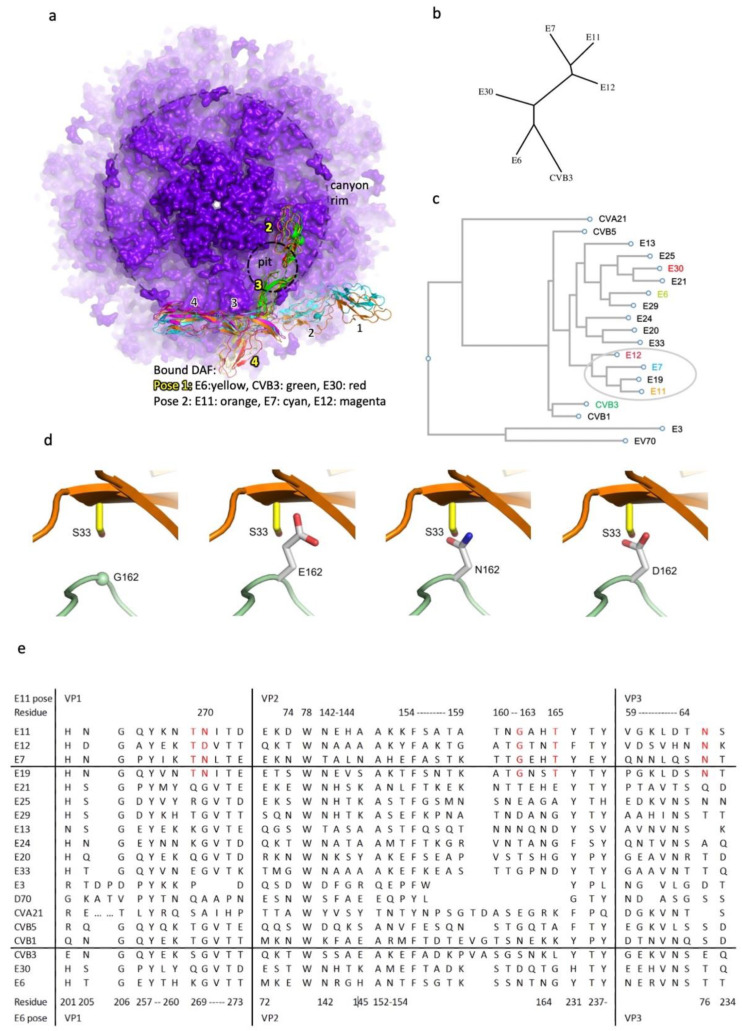
Binding mode of DAF. (**a**) Two different binding patterns of DAF with different enteroviruses. The viral capsid is shown as surface representation, one pit region in the canyon region of the capsid is marked by a dashed circle and the outer rim of the canyon is marked by a larger dashed circle. DAF molecules are shown in cartoon representation with different colors indicating the different viruses they bound with (green: CVB3, PDB: 7VY5; yellow: E6, PDB:6ILK; red: E30, PDB:7C9W; cyan: E7, PDB:3IYP; orange: E11, structure in this paper; magenta: E12, PDB:1UPN). (**b**) Phylogenetic tree of 6 DAF binding enteroviruses based on the capsid structures. (**c**) Phylogenetic tree of 20 DAF-binding enteroviruses based on the capsid sequences (colours as in (a)). (**d**) Substitutions of VP2 G162 (**left** panel in green) conserved in viruses with E11 DAF binding mode with D/N/E (second to forth panel, respectively) found in viruses with E6 DAF binding mode would clash with S33 of DAF (orange). (**e**) Gene sequence alignment of 20 DAF-binding enteroviruses. Key residues for the E11 and E6 interactions are listed above and below the sequence alignment, respectively and those marked in red are conserved among E11 pose enteroviruses and E19.

**Table 1 viruses-14-02625-t001:** Data collection, reconstruction and model refinement.

Data collection		
Voltage (kV)	300	
Frames	24	
Dose rate (e^-^/Å ^2^/s)	7	
Total dose (e^-^/Å ^2^)	41	
Calibrated pixel size (Å)	1.05	
Defocus (μm)	–1 to –3	
**Reconstruction and model refinement**		
	Relion	Cryosparc ^*^
Movies	6469	7088
Particles	6394	201,358 ^&^
Map resolution (Å)	3.1	3.93
Map sharpening B-factor (Å^2^)	–178.1	–181.5
Model-to-map fit, CC_mask	0.88	0.54
R.m.s.d., bonds (Å)	0.003	0.003
R.m.s.d., angles (°)	0.55	0.581
All-atom Clash score	5.51	9.36
Rotamer outliers (%)	0.68	0.80
**Ramachandran plot**		
Favored (%)	97.57	98.01
Allowed (%)	2.43	1.95
Outliers (%)	0.00	0.04

* Cryosparc was used to refine DAF binding area. ^&^ I1 symmetry expansion was applied to particles, which increases the particle number 60 times.

**Table 2 viruses-14-02625-t002:** List of protein interactions between E11 and DAF.

Residue in E11	Residue in DAF	Distance (Å)
**Potential hydrogen bond**		
VP2 154 LYS	168 GLU	2.60
VP2 156 SER	191 ASN	3.05
VP2 159 ALA	192 THR	3.16
VP2 161 ASN	173 GLN	3.57
VP2 161 ASN	192 THR	3.66
**Potential salt bridge**		
VP2 154 LYS	168 GLU	3.52
**Potential hydrophobic interaction**		
VP2 159 ALA	191 ASN	3.93
VP2 163 ALA	175 ASP	2.74
VP2 165 THR	173 GLN	3.77
VP2 165 THR	175 ASP	3.74
VP3 59 VAL	248 SER	3.70
VP3 60 GLY	246 ARG	2.63

## Data Availability

The coordinates are deposited in the PDB with accession codes 8B8R, 8B9F. Electron density maps are available from EMDB with accession codes EMD-15920, EMD-15930.

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
