# Peer review of "Switching of Receptor Binding Poses between Closely Related Enteroviruses"

_viruses, 2022, doi:10.3390/v14122625_

Round 1
Reviewer 1 Report
Manuscript Zhou et al. is focused on to structural study of echovirus 11 (E11) complex with its receptor DAF using cryo-EM. Manuscript is well written and easy to understand for readers, however additional experimental data is required to verify the structure (the binding interface between SCRs 3 and 4 of DAF to E11 (VP2 and VP3)). Therefore, I would propose a major revision before the publication.
(1) Binding interface between SCRs 3 and 4 of DAF to E11 (VP2 and VP3) must be confirmed by measuring the binding affinity for WT and several mutants (by mutating amino acids which are involved in the protein-protein interactions). These additional data should be included in manuscript.
(2) To understand the role of glycan in complex formation and positioning of DAF, I would propose to do mutagenesis and check it effects on complex formation, to support the result as shown in Fig. 2D, glycan attached to residue N95 of DAF (shown as a magenta circle) and residue 73 (amino acid?) of VP2 (marked with a green circle).
(3) Line 297 and 298, Include both PDB and EMBD accession code(s) XXX ?
Reviewer 2 Report
In this paper, the authors reported the cryoEM structures of echovirus 11 (E11) virions complexed with a host receptor, DAF. The key residues that were involved in the interactions were identified and an N-linked glycan of DAF was also found to contribute to the interaction. The binding pattern of DAF on E11 mimics the binding poses of E7 and E12, which also shares specific residues on VP1-3. However, the E11 binding pose is different from the E6 pose, indicating distinct evolution paths of two clades of the viruses.
1. Did you test or was there a previous study that reported the binding affinity of DAF with VP2/VP3?
2. Figure 1d&f: it would be nice to show the binding pattern of DAF on a complete virion (like highlighting DAF in 1e and zooming in for details in 1f,g,&h).
3. Figure 1g: I wonder which terminus of DAF is associated with the cell membrane, please label it in the figure.
4. Figure 2c: Please add dashes between interacted residues listed in Table 2 (like K154-E168).
5. Figure 2d: can you try to build the N-linked glycan and the side chain(s) it interacts with on the virus in the map? Are there similar glycan interactions in the structures of E7/E12 virions?
6. Please update the PDB and EMDB ID for the structures in this paper.
7. Please add the workflow for cryoEM data processing as a supplementary figure.
8. Please describe how the phylogenic trees (figure 3 b&c) were made in the methods.
Round 2
Reviewer 1 Report
As per author’s response1 “Mutational analysis would not signifiantly add to the paper since the complex structure itself confirms the binding interface which has indeed been reported previously.”
I would suggest updating the discussion with more details, to correlate current finding with those reported previous studies.
Author Response
Thanks. We have expanded the discussion to make it clearer and correlate our results with previous affinity and mutational analyses.